# Characterization of Phosphate Compounds along a Catena from Arable and Wetland Soil to Sediments in a Baltic Sea lagoon

Julia Prüter [1], Rhena Schumann [2] , Wantana Klysubun [3] and Peter Leinweber [1,4,]*

1   Soil Science, Faculty of Agricultural and Environmental Science, University of Rostock, Justus-von-Liebig-Weg 6, 18059 Rostock, Germany
2   Department of Applied Ecology, Institute of Biological Sciences, University of Rostock, Albert-Einstein-Straße 3, 18059 Rostock, Germany
3   Synchrotron Light Research Institute, Muang District, 111 University Avenue, Nakhon Ratchasima 3000, Thailand
4   Department of Life, Light and Matter, Interdisciplinary Faculty, University of Rostock, Albert-Einstein-Straße 25, 18051 Rostock, Germany
*   Correspondence: peter.leinweber@uni-rostock.de

**Abstract:** Phosphorus (P) is an indispensable nutrient for arable crops, but at the same time, contributes to excessive eutrophication in aquatic ecosystems. Knowledge about P is essential to assess the possible risks of P being transported towards vulnerable aquatic ecosystems. Our objective was to characterize P along a catena from arable and wetland soils towards aquatic sediments of a shallow lagoon of the Baltic Sea. The characterization of P in soil and sediment samples included a modified sequential P fractionation and P *K*-edge X-ray absorption near edge structure (XANES) spectroscopy. The concentrations of total P ranged between 390 and 430 mg kg$^{-1}$ in the arable soils, between 728 and 2258 mg kg$^{-1}$ in wetland soils and between 132 and 602 mg kg$^{-1}$ in lagoon sediments. Generally, two sinks for P were revealed along the catena. The wetland soil trapped moderately stable P, Al-P and molybdate-unreactive P (MUP), which are most likely organically bound phosphates. Sediments at the deepest position of the catena acted as a sink for, MUP compounds among the lagoon sediments. Thus, wetlands formed by reed belts can help to prevent the direct transfer of P from arable soils to adjacent waters and deeper basins and help to avoid excessive eutrophication in shallow aquatic ecosystems.

**Keywords:** phosphorus; gradient; sequential fractionation; XANES spectroscopy





## 1. Introduction

Coastal wetlands, as open systems, link terrestrial and aquatic ecosystems and play an important role in the environment as habitats for fish and birds, for erosion protection and for the biogeochemical cycling of nutrients in nearby coastal sites or adjacent land [1–3]. Phosphorus (P) is an essential element for all living organisms, in coastal wetlands as well as in terrestrial ecosystems, as it contributes to crop yields in arable farming, but at the same time, it has been identified as one of the major factors responsible for eutrophication in wetlands and aquatic ecosystems [3,4]. Eutrophication causes various negative environmental impacts, such as excessive algal blooms, water oxygen depletion and the release of hazardous toxins [4,5]. As P from agricultural fertilization is primarily conserved in soils and can be transported from terrestrial to aquatic ecosystems [6], it is important to develop sustainable agricultural practices and similarly ensure the protection of the environment [7].

The German coast of the Baltic Sea is characterized by large coastal wetlands, adjoining arable soils on one side and aquatic lagoon systems on the other side [8]. Thus, there is a high probability of nutrients, and particularly P, from fertilized arable soils being transported into the adjacent wetland soils, water bodies and sediments, but these expected transfer processes have not been disclosed in detail. Generally, the speciation of P affects

the risk of P transportation to surface waters [9], as well as P availability for plant uptake. Thus, knowledge about the P speciation in soils and its transformation processes towards the coast are essential to assess the possible risks of P transportation into aquatic ecosystems and to develop measures to prevent excessive P inputs into vulnerable water bodies.

Many previous studies have investigated chemical P composition in agricultural soils (e.g., [10–12]), sediments (e.g., [13–15]) and P in the water column (e.g., [16–18]). Nonetheless, in-depth knowledge at the molecular level about P speciation and the transport of P species from soils to sediments is scarce [19]. Furthermore, few studies have investigated the P speciation of samples at the fluent boundaries between terrestrial soils and sediments in aquatic environments along sequences. For instance, accumulations of organic P ($P_o$) in muck soils and fractions of inorganic P ($P_i$) in adjacent river/lake sediments have been reported from Ontario, Canada [20]. Another investigation of a transect from arable soils in northern Germany towards sediments of the central Baltic Sea revealed a similar distribution of $P_i$ and $P_o$ fractions in the soils and sediments [21]. Furthermore, an increase in the proportion of stable P fractions (i.e., $H_2SO_4$-P and residual-P) compared to iron-associated P with increasing distances was reported in the same study, which covered a transect from the coastline to central Baltic Sea sediments with a length of about 600,000 m [21]. However, it is unknown which P transformation processes occur at a smaller scale from coastal arable and wetland soils to adjacent sediments from a shallow Bodden of the Baltic Sea with a transect length of about 700 m.

The aim of this study was to characterize the P compounds along a sequence from arable and wetland soils towards aquatic sediments from a shallow lagoon of the Baltic Sea to fill in the knowledge gap in the course of medium-scale (hundreds of meters) spatial expansion. We want to test whether the previously disclosed transition of labile Al- and Fe-associated P species in terrestrial soils to more stable and Ca-bound P in aquatic sediments along a large-scale transect towards the Central Baltic Sea [21] also applies to the specific geomorphological setting of a medium-scale transect. The overall objective of the research is to disclose the general pathways of P species transformations along moisture gradients from land to waters, irrespective of the scale of the study area.

## 2. Materials and Methods

### 2.1. Sampling Area, Soil and Sediment Collection

Sampling of the soils and sediments took place in summer 2018 near the village Dabitz in Mecklenburg-Western Pomerania in the area of Darss-Zingst Bodden Chain, a lagoon system at the Southern Baltic Sea in Germany. The transect expands from N54°22′08.00″ E12°48′08.00″ to N54°22′09.31″ E12°48′33.50″. The study area includes three ecologically different sites: (1) an arable field cropped with barley; (2) an adjoining coastal wetland covered by *Phragmites australis*; and (3) a shallow water body with a mean water depth of 2 m. For more details, such as a description of the vegetation, water and sediment characteristics of the wetland, see an earlier investigation [22].

The soil samples and sediment cores were taken at two depths along a transect from an arable field (A1, A2) continued to the directly bordering wetland (W1, W2) and the adjacent Bodden sediments at three different water depths (S1, S2; S3, S4; and S5, S6). The study area and the sampling sites along the transect are visualized in Supplemental Figure S1. The sample description and labeling are compiled in Table 1, and Figure 1 shows a schematic drawing of the sampling strategy. The total distance between the first and the terminal sample of the transect is approximately 700 m. Two or three single subsamples per depth (decided according to visual agreement/disagreement in sample color/structure) were taken at an area of approximately one m² and merged to one mixed sample for each soil and sediment sample location and depth. This merging was conducted if samples showed almost similar results for the basic parameters, such as water content (differing by 0 to 3% (*w/w*) except for two outliers with 13 and 17% (*w/w*) for samples W1 and W2), pH (differing by 0.1 to 0.8 pH-units), and C, N and S contents (differing by 0 to 1% (*w/w*), except for one outlier with 8% difference in C-content (sample W1)). In this way, we

reduced the effects of small-scale heterogeneity and, at the same time, limited the number of samples to be analyzed. Furthermore, in a recent study examining sample pretreatments of soils and sediments, we concluded that an identical pretreatment of the soils and sediments resulted in no fundamental changes to their P speciation [23]. For this reason, all of the soil and sediment samples of the present investigation were dried at 40 °C, sieved <2 mm and finely ground in a mortar mill prior to further analyses.

**Table 1.** Label, sample type, origin, sampling depth and coordinates of the collected soil and sediment samples.

| Label | Sample Type | Origin | Sampling Depth in cm | Coordinates |
|---|---|---|---|---|
| A1 | Soil | arable field | 0–30 | N 54°22′08.00″ |
| A2 | Soil | arable field | 30–60 | E 12°48′08.00″ |
| W1 | Soil | Wetland | 0–10 | N 54°22′08.30″ |
| W2 | Soil | Wetland | 30–50 | E 12°48′12.60″ |
| S1 | Sediment | water depth 52 cm | 0–5 | N 54°22′09.00″ |
| S2 | Sediment | water depth 52 cm | 5–10 | E 12°48′17.60″ |
| S3 | Sediment | water depth 63 cm | 0–5 | N 54°22′09.20″ |
| S4 | Sediment | water depth 63 cm | 5–10 | E 12°48′20.00″ |
| S5 | Sediment | water depth 230 cm | 0–5 | N 54°22′09.31″ |
| S6 | Sediment | water depth 230 cm | 5–10 | E 12°48′33.50″ |

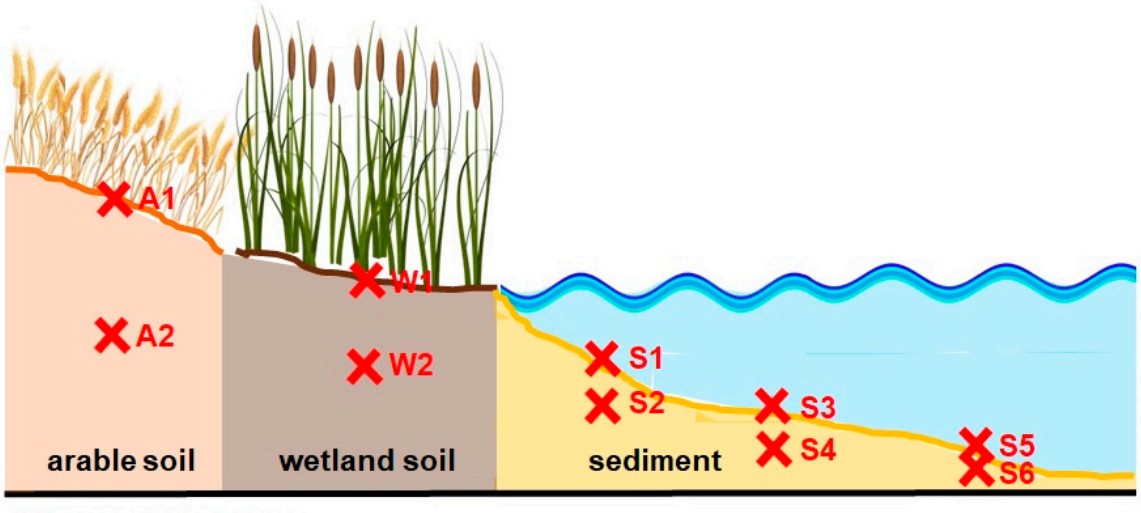

**Figure 1.** Schematic illustration of the soil and sediment sampling spots at the study site in Dabitz.

The soil texture of the cropland at the study site was characterized as loamy sand [22] and the sediment textures of the adjacent Bodden sediments were fine to medium sands [24].

## 2.2. Determination of Water Contents and the Total Concentrations of C, N, S, CaCO₃, P, Ca, Mg, Al, Fe

The water contents were determined by sample weighing before and after drying at 105 °C to constant weight. The contents of total carbon (C), nitrogen (N) and sulphur (S) were obtained through the dry combustion of finely ground soil and sediment material using an elemental analyzer (VARIO EL, Elementar Analysensysteme GmbH, Hanau, Germany). The percentages of soil and sediment calcium carbonate ($CaCO_3$) were determined using a Scheibler calcimeter, by calculating the carbon dioxide ($CO_2$) volume resulting from the reaction of hydrochloric acid (HCl) with the sample $CaCO_3$. The elemental concentrations of total P ($P_t$), calcium (Ca), magnesium (Mg), aluminum (Al), iron (Fe) and

zinc (Zn) were determined by microwave-assisted digestion (Mars Xpress CEM GmbH Kamp-Linfort, Germany) of $\leq$50 mg soil or sediment with *aqua regia* consisting of 2 mL nitric acid ($HNO_3$) and 6 mL HCl (ISO standard 11466). The element concentrations in the digests were determined with an inductively coupled plasma-optical emission spectrometer (ICP-OES) at wavelengths of 214.914 nm for P, 317.933 nm for Ca, 258.213 nm for Mg, 396.153 nm for Al and 238.204 nm for Fe.

### 2.3. Sequential P Fractionation

A slightly modified sequential P-fractionation method was used to extract different P fractions from soil and sediment [25,26]. Approximately 0.42 g finely-ground soil or sediment was weighed into 50 mL centrifuge tubes. The samples were shaken for 18 h at room temperature, followed by centrifugation at $4000\times$ *g* for 20 min, and decanted. The chemical P fractionation included the following extraction steps: (1) $H_2O$; (2) anion resin strips (55,164 2S, BDH Laboratory Supplies, Poole, England); (3) 0.5 M $NaHCO_3$; (4) 0.1 M NaOH; (5) 1 M $H_2SO_4$. In the second extraction step, P was removed from the resin using 1 M HCl. The P fractions were interpreted as follows: water ($H_2O$-P) and resin P (resin-P) representing the easily exchangeable and mobile P, molybdate-reactive ($NaHCO_3$-$P_{mr}$) and molybdate-unreactive ($NaHCO_3$-$P_{mu}$) bicarbonate P representing labile $P_{mr}$ and $P_{mu}$ weakly adsorbed to mineral surfaces as well as microbial P, molybdate-reactive (NaOH-$P_{mr}$) and molybdate-unreactive (NaOH-$P_{mu}$) sodium hydroxide P representing moderately labile $P_{mr}$ and $P_{mu}$ adsorbed to Al- and Fe-oxide minerals and P in humic substances, eventually also associated with Fe and Al. The $H_2SO_4$-P fraction represents the insoluble P, associated with Ca and Mg minerals and apatite [25]. The $P_t$ in the different extracts was measured in the decanted supernatants using an ICP-OES, while the remaining sediment pellet was used for the next extraction step. The molybdate-reactive P ($P_{mr}$) concentrations in the extracts were determined colorimetrically with the molybdate blue method [27]. The concentration of the molybdate-unreactive P ($P_{mu}$) was estimated by subtracting $P_{mr}$ from $P_t$. The concentration of the non-extracted P (Residual-P) was calculated as the difference between the sum of the P fractions and the $P_t$ concentration determined after digestion with *aqua regia*.

The application of the fractionation scheme, which was originally developed for estimating the plant availability of P in amended soils, to sediments is a challenge, as different fractionation schemes have been developed and applied for sediments (e.g., [28]). In the sediment method, samples are sequentially extracted with $NH_4Cl$, dithionite-citrate-bicarbonate, NaOH and HCl. It is similar to the soil fractionation in starting with mild extractant to remove loosely sorbed P ($NH_4Cl$ vs. $H_2O$ and anion exchange resin), P bound in humic substances (NaOH) and finalizing with the removal of relatively stable Ca-bound P by strong mineral acid (concentrated HCl or $H_2SO_4$). The major difference is the reduction in the Fe-oxides by dithionite-citrate-bicarbonate in the sediment fractionation, which is not involved in soil fractionation. Thus, the latter fractionation does not allow for estimating the amount of Fe-bound P; however, this can be derived from P *K*-edge XANES in a multimethod approach. Thus, in summary, the P fractionation schemes for soils and sediments are expected to yield similar results for the proportions of the most labile and most stable P fractions, and may be replaced by each other in this respect.

### 2.4. P K-Edge XANES Analysis

The P *K*-edge XANES spectra were recorded at the Synchrotron Light Research Institute (SLRI) in Nakhon 65 Ratchasima, Thailand, on the beamline 8 (BL8) of the electron storage ring with a covering photon energy from 1.25 to 10 KeV, electron energy operated at 1.2 GeV and a beam current of 80–150 mA [29]. The XANES data were collected from dry and finely-ground samples, thinly spread on P-free kapton tape (Lanmar Inc., Northbrook, IL, USA) and attached to a plastic sample holder. The data collection was operated in standard conditions with energy calibration by standard elemental P and allocating the reference energy ($E_0$) at 2145.5 eV using the maximum peak of the first derivative spectrum.

All of the spectra were recorded at photon energies between 2045.5 and 2495.5 eV in step sizes of 5 eV (2045.5 to 2105.5 eV and 2245.5 to 2495.5 eV), 1 eV (2105.5 to 2135.5 eV and 2195.5 to 2245.5 eV) and 0.25 eV (2135.5 to 2195.5 eV) with a 13-channel germanium detector in fluorescence mode. Two to four scans were collected and averaged for each sample.

All of the P $K$-edge XANES spectra were normalized, and the replicates were merged. Linear combination fitting (LCF) was performed using the ATHENA software package [30] in the energy range between $-20$ eV and $+30$ eV of $E_0$. The XANES spectral data were baseline corrected in the pre-edge region between 2115 and 2145 eV and normalized in the post-edge region of 2190–2215 eV. The same ranges were used for the reference P $K$-edge XANES spectra to achieve consistency in the following fitting analysis [31]. To achieve the best compatible set of references with each specified sample spectrum, LCF analysis was performed using the combinatorics function of the ATHENA software to attain all of the possible binary to quaternary combinations between all 19 P reference spectra, in which the share of each compound was $\geq 10\%$. The following set of reference P $K$-edge XANES spectra, all recorded in SLRI under the same adjustments [31,32], were used for fitting and calculations: Ca, Al and Fe phytate, noncrystalline and crystalline $AlPO_4$, noncrystalline and crystalline $FePO_4 \cdot 2H_2O$, Ca 5-hydroxyapatite ($Ca_5(OH)(PO_4)_3$), inositol hexakisphosphate/phytate (IHP), ferrihydrite–IHP, montmorillonite–Al–IHP, soil organic matter (OM) Al–IHP (SOM–Al–IHP), ferrihydrite–orthophosphate, montmorillonite–Al–orthophosphate, SOM–Al–orthophosphate, boehmite–IHP, boehmite–10 orthophosphate, $CaHPO_4$, $Ca(H_2PO_4)_2$ and $MgHPO_4$. The $R$-factor values were used as goodness-of-fit criteria and significant differences between fits were evaluated using the Hamilton test ($p < 0.05$) [33], with the number of independent data points calculated by ATHENA and estimated as the data range divided by the core-hole lifetime broadening. The best fits of the P reference compound combinations were considered as the most probable P species in the material. If the $R$ factors of the fits with the same number of reference compounds were not significantly different from each other according to the Hamilton test, the fit proportions were averaged. For this reason, the averaged proportions of some reference compounds can be $\leq 10\%$.

### 2.5. Statistical Analysis

The data analysis was performed using the open-source statistical software R (version 3.6.3). The R package agricolae was used for the Tukey's honest significant difference (HSD) test, which enables multiple comparisons (significance level 0.05). The Tukey's HSD test was used to find differences in the elemental concentrations and the results of the sequential P fractionation between sampling sites and depths.

## 3. Results

### 3.1. Soil and Sediment Characteristics

The water contents of the arable and wetland soils and sediments differed greatly. The arable soil samples A1 and A2 contained less than 10% ($w/w$) water, while the wetland soils W1 and W2 contained up to 62% ($w/w$) water. The sediments S1 to S4 had about 20% ($w/w$) water and maximum water contents of up to 75% were measured in the sediments S5 and S6 (Table 2). $CaCO_3$ was found in the arable soil samples A1 (6%) and A2 (14%). The wetland soil and the sediments did not contain measurable amounts of $CaCO_3$.

The concentrations of the total C ranged between 14,500 and 136,900 mg kg$^{-1}$ in the soil and between 2500 and 67,000 mg kg$^{-1}$ in the sediment samples (Table 2). Whereas the C contents were extremely low in the sediments S1 to S4, there were much higher contents in the sediments S5 and S6 from a water depth of 230 cm. The average C percentages were more than two times higher in the arable and wetland soils compared to the sediments. The maximum N concentrations were determined in the soil W1 (12,500 mg kg$^{-1}$) and in the sediments S5 (7500 mg kg$^{-1}$) and S6 (6200 mg kg$^{-1}$). The average N concentrations were, again, about two times higher in the soils compared to the sediments. The highest S

concentrations were present in the sediments S5 (15,600 mg kg$^{-1}$) and S6 (14,700 mg kg$^{-1}$), while all of the other sediment and soil samples had <4000 mg kg$^{-1}$ S.

**Table 2.** Average proportions of water in % and elemental concentrations of carbon (C), nitrogen (N) and sulphur (S), *n* = 2; and of phosphorus (P), calcium (Ca), magnesium (Mg), aluminum (Al), and iron (Fe) in mg kg$^{-1}$ and their ratios (C/P, P/Ca, P/Mg, P/Al, P/Fe) determined by ICP-OES; *n* = 3 in the upper and lower soil and sediment samples. Significant differences at 5% probability level between samples are designated by different letters (a, b, c, d, e, f).

| Sample | Water Content | CaCO$_3$ | C | N | S | P | Ca | P/Ca | Mg | P/Mg | Al | P/Al | Fe | P/Fe |
|---|---|---|---|---|---|---|---|---|---|---|---|---|---|---|
| | % | % | mg kg$^{-1}$ | | | | mg kg$^{-1}$ | | mg kg$^{-1}$ | | mg kg$^{-1}$ | | mg kg$^{-1}$ | |
| A1 | 05 | 6 | 14,583 | 842 | 375 | 430 [e] | 27,835 [b] | 0.02 | 2974 [c] | 0.14 | 10,982 [bx] | 0.04 | 13,738 [e] | 0.03 |
| A2 | 09 | 14 | 24,400 | 272 | 327 | 390 [e] | 72,935 [a] | 0.01 | 4134 [b] | 0.09 | 10,768 [bc] | 0.04 | 13,603 [e] | 0.03 |
| W1 | 62 | 0 | 136,917 | 12,580 | 3648 | 2258 [a] | 6944 [c] | 0.33 | 4869 [a] | 0.46 | 18,380 [ax] | 0.12 | 34,382 [a] | 0.07 |
| W2 | 46 | 0 | 48,683 | 4578 | 1963 | 728 [b] | 3589 [d] | 0.20 | 3002 [c] | 0.24 | 11,344 [bx] | 0.06 | 15,056 [d] | 0.05 |
| S1 | 20 | 0 | 2600 | 345 | 525 | 140 [f] | 909 [e] | 0.15 | 325 [d] | 0.43 | 737 [dx] | 0.19 | 861 [f] | 0.16 |
| S2 | 20 | 0 | 2900 | 375 | 560 | 137 [f] | 1173 [e] | 0.12 | 404 [d] | 0.34 | 913 [dx] | 0.15 | 1104 [f] | 0.12 |
| S3 | 21 | 0 | 2550 | 340 | 475 | 132 [f] | 639 [e] | 0.21 | 289 [d] | 0.46 | 681 [dx] | 0.19 | 847 [f] | 0.16 |
| S4 | 20 | 0 | 2450 | 315 | 450 | 135 [f] | 445 [e] | 0.30 | 285 [d] | 0.47 | 750 [dx] | 0.18 | 1018 [f] | 0.13 |
| S5 | 65 | 0 | 67,050 | 7475 | 15,650 | 602 [c] | 4839 [d] | 0.12 | 4830 [a] | 0.12 | 9940 [cx] | 0.06 | 17,637 [c] | 0.03 |
| S6 | 75 | 0 | 59,550 | 6190 | 14,740 | 551 [d] | 3652 [d] | 0.15 | 4751 [a] | 0.12 | 10,489 [bc] | 0.05 | 18,452 [b] | 0.03 |

The order of the concentrations of P agreed with the C and N concentrations of the soils and sediments (Table 2). The soil W1 had the maximum concentration, of 2258 mg P kg$^{-1}$. Minimum concentrations of 130 to 140 mg P kg$^{-1}$ were present in the sediments S1 to S4, with no significant differences, whereas sediments S5 and S6 again had higher P concentrations, of 602 and 551 mg kg$^{-1}$. On average, the P concentrations were more than three times higher in the arable and wetland soils than in the sediments. The highest Ca concentration was present in the soil A2 and, overall, the average Ca concentrations were about 14 times higher in the soils compared to the sediments. For the concentrations of the elements Mg, Al and Fe, there were also differences between the soils and sediments, but not as clear as for Ca. The arable and wetland soils contained, on average, about 1.8 to 3.3 times more Mg, Al and Fe than the sediments.

### 3.2. Sequentially Extracted P Fractions

The sequential chemical fractionation extracted, on average, 89% of the P$_t$ across all of the samples (Table 3). With the exception of W1, H$_2$SO$_4$-P$_{mr}$ was generally the largest fraction, ranging between 25% and 67% of P$_t$ in the soil and sediment samples. The maximum amount of P in the wetland soil W1 was present in the residual-P fraction. In most of the sequentially extracted fractions, the proportions of P$_{mr}$ were higher than P$_{mu}$, except for the sediments S5 and S6 in the fractions of NaHCO$_3$ and NaOH, and for the soil W2 in the fraction of NaOH, where P$_{mu}$ was predominant. Furthermore, it is noticeable that, although the absolute concentrations of P in the easily exchangeable and plant-available fractions of H$_2$O-P and resin-P were rather low in the sediments S1 to S4, their relative amounts of 7% to 24% of P$_t$ were higher than in the soils and deeper sediments. There were no significant differences among the concentrations of NaHCO$_3$-P$_{mr}$ among the soils and sediments, except for W1, in which the maximum concentration of 275 mg P kg$^{-1}$ was determined. In the fraction of NaOH-P$_{mr}$ there were also few significant differences between the samples. Exclusively the upper and lower wetland soil samples W1 and W2 contained significantly more NaOH-P$_{mr}$ than the other soils and sediments.

**Table 3.** Concentrations (mg kg$^{-1}$) and percentages (%) of the sequentially extracted molybdate-reactive ($P_{mr}$) and molybdate-unreactive ($P_{mu}$) P fractions $H_2O$-P, resin-P, $NaHCO_3$-P, NaOH-P, $H_2SO_4$-P, and residual-P, of total P ($P_t$) and the sums of $P_{mr}$ and $P_{mu}$ determined in the soil and sediment samples. Significant differences at 5% probability level between samples are designated by different letters (a, b, c, d, e, f), $n = 3$.

| Sample | $H_2O$-$P_{mr}$ mg kg$^{-1}$ | (%) | $H_2O$-$P_{mu}$ mg kg$^{-1}$ | (%) | Resin-$P_{mr}$ mg kg$^{-1}$ | (%) | Resin-$P_{mu}$ mg kg$^{-1}$ | (%) | $NaHCO_3$-$P_{mr}$ mg kg$^{-1}$ | (%) | $NaHCO_3$-$P_{mu}$ mg kg$^{-1}$ | (%) | NaOH-$P_{mr}$ mg kg$^{-1}$ | (%) | NaOH-$P_{mu}$ mg kg$^{-1}$ | (%) | $H_2SO_4$-$P_{mr}$ mg kg$^{-1}$ | (%) | $H_2SO_4$-$P_{mu}$ mg kg$^{-1}$ | (%) | Residual-P mg kg$^{-1}$ | (%) | $P_t$ mg kg$^{-1}$ | Sum $P_{mr}$ mg kg$^{-1}$ | (%) | Sum $P_{mu}$ mg kg$^{-1}$ | (%) |
|---|---|---|---|---|---|---|---|---|---|---|---|---|---|---|---|---|---|---|---|---|---|---|---|---|---|---|---|
| A1 | 13 c | (3) | 2 cx | (0) | 18 dex | (4) | 1 a | (0) | 15 b | (3) | 13 de | (3) | 12 c | (3) | 14 bx | (3) | 222 bx | (52) | 37 ab | (9) | 83 bcd | (19) | 430 e | 280 | (65) | 67 | (16) |
| A2 | 9 c | (2) | 0 cx | (0) | 11 exx | (3) | 0 a | (0) | 47 b | (12) | 2 ex | (1) | 9 c | (2) | 0 bx | (0) | 261 bx | (67) | 29 ab | (7) | 22 cdx | (6) | 390 e | 337 | (86) | 31 | (8) |
| W1 | 26 a | (1) | 23 ax | (1) | 143 axx | (6) | 6 a | (0) | 275 a | (12) | 162 ax | (7) | 443 a | (20) | 156 ax | (7) | 401 ax | (18) | 116 ax | (5) | 505 axx | (22) | 2258 a | 1288 | (57) | 464 | (21) |
| W2 | 26 a | (4) | 8 bc | (1) | 40 bcd | (5) | 0 a | (0) | 50 b | (7) | 51 bx | (7) | 75 b | (10) | 92 ab | (13) | 183 bc | (25) | 55 ab | (8) | 149 bxx | (20) | 728 b | 374 | (51) | 206 | (28) |
| S1 | 14 c | (10) | 7 bc | (5) | 26 cde | (18) | 6 a | (4) | 20 b | (14) | 1 ex | (1) | 9 c | (6) | 0 bx | (0) | 49 cd | (35) | 14 bx | (10) | 0 dxx | (0) | 140 f | 118 | (81) | 28 | (19) |
| S2 | 13 c | (9) | 10 bc | (7) | 33 cde | (24) | 4 a | (3) | 15 b | (11) | 1 ex | (1) | 9 c | (6) | 0 bx | (0) | 53 cd | (39) | 15 bx | (11) | 0 dxx | (0) | 137 f | 122 | (81) | 29 | (19) |
| S3 | 27 a | (21) | 16 ab | (12) | 18 dex | (14) | 0 a | (0) | 7 b | (5) | 1 ex | (1) | 9 c | (7) | 0 bx | (0) | 44 dx | (34) | 10 bx | (8) | 0 dxx | (0) | 132 f | 105 | (79) | 28 | (21) |
| S4 | 19 b | (14) | 10 bc | (7) | 17 dex | (13) | 0 a | (0) | 15 b | (11) | 1 ex | (1) | 11 c | (8) | 0 bx | (0) | 60 cd | (45) | 11 bx | (8) | 0 dxx | (0) | 135 f | 123 | (85) | 22 | (15) |
| S5 | 20 b | (3) | 15 ab | (2) | 57 bxx | (9) | 4 a | (1) | 8 b | (1) | 42 bc | (7) | 18 c | (3) | 57 ab | (9) | 265 bx | (44) | 0 bx | (0) | 115 bcx | (19) | 602 c | 368 | (61) | 119 | (20) |
| S6 | 20 b | (4) | 17 ab | (3) | 47 bcx | (9) | 3 a | (1) | 3 b | (1) | 27 cd | (5) | 12 c | (2) | 35 bx | (6) | 199 bx | (36) | 77 ab | (14) | 111 bcd | (20) | 551 d | 282 | (51) | 158 | (29) |

### 3.3. Bulk P K-Edge XANES Spectra

All XANES spectra and *R* factors from LCF of XANES analyses are compiled in Supplementary Materials. The XANES spectra were all characterized by an intense white line peak at around 2152 eV and varying pre- and post-edge features. The *R* factors from the LCF were 0.0033 to 0.0161 for the soil samples and 0.0022 to 0.0095 for the sediment samples (all *R* factors are compiled in Table S1 of Supplementary Materials). The P speciation of all the samples, based on the XANES spectra and LCF, are displayed in Figure 2 (corresponding XANES spectra in Figure S2 of Supplementary Materials). The proportions of the specific Fe-, Al- and Ca-P compounds were grouped to Fe-P, Al-P, Ca-P, Mg-P and $P_o$. The average proportions of the summed Ca-P compounds were lower in the arable and wetland soils (38%) compared to the sediments (89%). All of the arable soil and sediment samples were dominated by Ca-associated P compounds. In the upper wetland soil, W1, exclusively Al-P compounds were assigned by the XANES spectroscopy, and in the corresponding subsoil sample W2, Al-, Fe-P and $P_o$ compounds were predominant. The proportions of $P_o$ compounds were exclusively present in A1, W2 and S5 in considerable amounts.

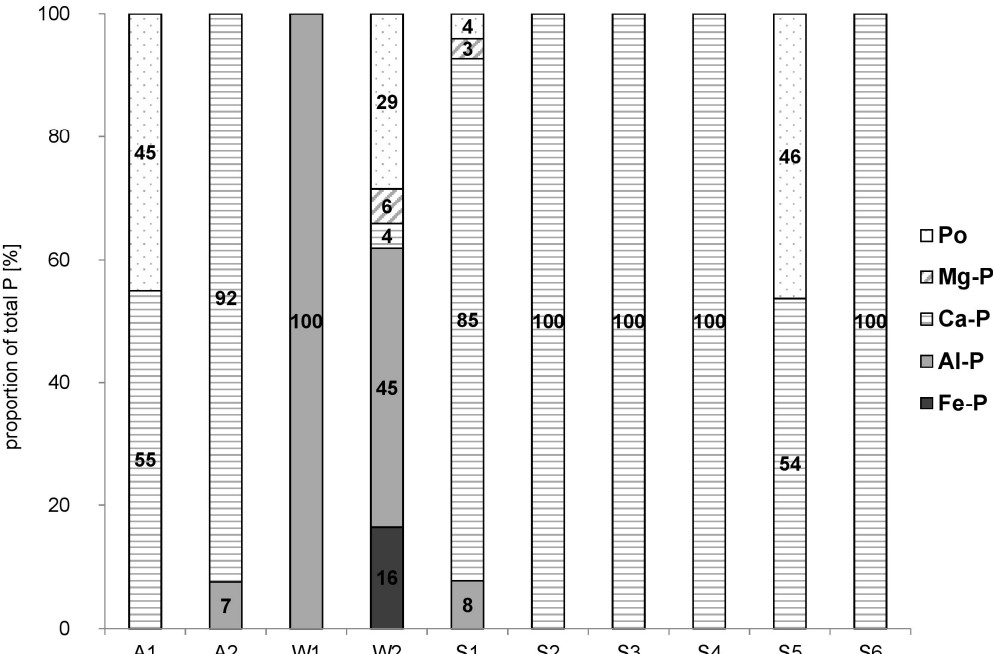

**Figure 2.** Proportions of P compounds as obtained by linear combination fitting (LCF) on P *K*-edge XANES spectra of upper and lower arable soil (A), wetland soil (W) and sediment samples (S).

## 4. Discussion

### 4.1. Elemental Characteristics

The wetland is located near the ground- and surface-water, whereas the arable soil is situated at a higher altitude, further away from these water sources, and simultaneously located on a slope, which is affected by runoff at the surface, erosion and subsurface drainage (Figure 1). Furthermore, the sampling took place in early summer, after a long dry period, so the arable soil had not received any precipitation for several weeks. The origin of the 6% $CaCO_3$ in A1 and 14% in A2 most likely originates from the underlying parent material glacial till at the arable field, which may have been partially incorporated into the soil profile by tillage. As the arable field has a relatively high elevation and a sloping relief towards the coast (Figure 1), erosion during rain fall events can transport solid matter into the wetland [22]. Thus, the topsoil material may have been transported away from the slope by erosion and, thus, the $CaCO_3$-containing underlying parent material may have been incorporated into the remaining topsoil by agricultural tillage. This explains the

higher proportion of $CaCO_3$ at a depth of 30–60 cm compared to 0–30 cm of the arable soil (Table 2), although regular liming may have added some $CaCO_3$ to the tilled soil layer.

With the exception of Ca, the hypothesized effect of erosion is reflected by the concentrations of all of the elements determined. We measured higher concentrations of P, Mg, Al and Fe in W1, the upper sample of wetland soil, than in the arable soil samples A1 and A2 (Table 2). The amount of Ca is significantly higher in the arable soil compared to the wetland soil because of the entry of $CaCO_3$ from the underlying parent material into the tilled soil, as already mentioned above.

The concentrations of $P_t$ were significantly higher in the topsoil of the wetland (2258 mg kg$^{-1}$) compared to the topsoil at the arable site (430 mg kg$^{-1}$) (Table 2); this indicates an accumulation of P compounds in the wetland. While P can be consumed by crops on the agricultural soil and transported away from the location with the harvest and by erosion due to the sloping relief, there is no cultivation with harvest and less sloping at the wetland soil, and restricted organic matter oxidation. Together, these factors facilitate an accumulation of organic matter as peat and of P stored in that substrate. The arable field has been used for farming since approximately 1945 [22]. At present, the field crops oil seed rape, wheat and barley, with high fertilizer demands, are cultivated [22]. Thus, it is likely that fertilizer P, in particular, is present in the arable soil and transported towards the wetland soil, although since the year 2000, only cow manure has been applied, rather than mineral P fertilizers [22]. This manure P seems to not or only little be transported beyond the wetland soil into the directly adjacent Bodden sediments, because the P concentrations in S1 to S4 were in a very low range, between 132 and 140 mg kg $^{-1}$ (Table 2). Thus, most of the nutrients, and particularly P, were transported from the arable soil towards the wetland soil and seem to be accumulated there for a longer time period. Similar relationships between the samples have been observed for the sediments S1 to S6. The amounts of P, Ca, Mg, Al and Fe were significantly higher in the sediments S5 and S6 from a greater water depth compared to most of the sediments from more shallow areas (S1 to S4) (Table 2).

*4.2. Sequential Chemical P Fractionation*

The concentrations of $P_t$ in the soils and sediments were in a similar range of $P_t$ contents in sediments from shallow lakes in the Yangtze River area in China [34] and, except for W1 and W2, they were below the $P_t$ contents, between 740 and 1230 mg P kg$^{-1}$, in bottom sediments of eutrophic lakes in central and western Poland [35]. The mean concentrations of $P_t$ were higher in the investigated soils compared to the adjacent sediments, agreeing with a study of transitional agricultural ecotones in Southern Germany, where concentrations of $P_t$ of the site average were also significantly higher than in the corresponding streambed sediment [36]. As the summed amounts of $P_{mu}$ were clearly higher in W1 and W2 compared to A1 and A2 (Table 2), this accumulation of $P_{mu}$ in the wetland soil can be attributed to an uptake of mainly $P_{mr}$ by the *Phragmites spp.* plants, its transformation to $P_{mu}$, likely primarily bound in organic matter, and its disposal as constituent of the wetland peat [37]. Furthermore, some $P_{mu}$ may have been transported from the arable site into the wetland, accumulating there. Although organic soils are known to have lower capacities for retaining excess P from fertilization and thus pose an increased risk of P loss to aquatic environments [38,39], the amounts of P were not particularly high in the directly adjacent sediments, S1 to S4 (Table 3). This could be due to high concentrations of stable fractions such as $H_2SO_4$-P and residual-P in the wetland, restricting processes of P mobilization at this site. Furthermore, a dilution of transported P within the Bodden water and/or sediments is possible or, alternatively, transferred P could have been mobilized immediately and accumulated by aquatic plant and animal organisms.

The proportions of $H_2O$-P and resin-P, characterized as labile P in soils [26] and P immediately available for uptake by phytoplankton [40], were lower in the arable and wetland soils (0–4%) compared to the sediments (0–24%) (Table 3). This pool of loosely bound P, similarly derived from a sediment-specific P-fractionation scheme after Psenner [28], is known to be seasonally variable in sediments, as well as being affected by enhanced

sedimentation and intensive degradation of OM during high summer temperatures [34,41]. Therefore, comparably large proportions in sediments are not implausible. The fraction of $NH_4Cl$-P, also characterized as mobile, exchangeable P, but determined with the sediment P fractionation method [28], included between 1 and 20% of $P_t$ in different sediments from eutrophic lakes in Poland [35]. Therefore, this fraction was in a similar range as the proportions of $H_2O$-P and resin-P fractions of the present study, pointing to the quantitation of similar pools sizes, irrespective of the differences in the fractionation schemes developed for soils and sediments.

Within the soil P fractionation scheme, P extracted by NaOH is interpreted as being predominantly associated with Al and Fe oxide minerals, which eventually combine with humic substances [25,26], whereas Fe-P compounds are separately estimated in the sediment-P-methods [28]. Particularly in the fractions of NaOH-$P_{mr}$ and NaOH-$P_{mu}$, higher P concentrations were measured in the wetland soil W1 and W2 than in the arable soil A1 and A2, and in the sediments S5 and S6 than in the other sediments from the Bodden (Table 3). Thus, the wetland soil and the sediments at the end of the transect at a water depth of 230 cm seem to be sinks for metal bound P and P in humic substances, which are likely combined with each other [42].

Acid extractable $P_{mr}$ accounted for the greatest proportion of $P_t$ in the sediments (up to 45%), which is in agreement with an investigation of lake sediments from [40]. This P fraction is known as stable, Ca-bound P [26], and is not readily available to phytoplankton [43]. An investigation of sediments from several eutrophic lakes in Poland also resulted in up to 45% acid extractable P (HCl-P) of $P_t$ [35], and in surface sediments of the Mediterranean Sea, 37% P of $P_t$ were determined in the fraction of Ca-bound P [44] with sediment fractionation methods [28]. The relative proportions of $H_2SO_4$-P were also very high in the arable soil compared to the wetland soil (Table 3). The available amounts of $CaCO_3$ in A1 and A2 (Table 2) can facilitate the formation of Ca-bound P at the location of the eroded arable slope compared to the wetland soil.

The statement that residual-P can constitute a significant proportion of $P_t$ from P fractionation in soils [45] is supported by the present study, as up to 22% of the $P_t$ in the soil samples was residual-P (Table 3). However, in the sediments, no residual-P was determined in S1 to S4, and approximately 20% was determined in the last two sediments, S5 and S6. The Phosphorus in the residual fraction was characterized as stable complexes with metal ions, pedogenic oxides or organic materials, such as lignin [39,46]. The sediments from more shallow areas contained more labile P and the sediments from a water depth of 230 cm, at the end of the transect under study, contained higher amounts of stable P, which is associated with Ca instead of metal ions, pedogenic oxides or complex organic materials (Table 3). In contrast to labile P fractions, such as resin-P and NaHCO3-P, the very stable P compounds within the fraction of residual-P pose a lower risk of P loss to the aquatic environment [47]. Thus, P can accumulate and be conserved in deep sediments at the end of the investigated transect.

### 4.3. P XANES Spectroscopy

The majority of the results of the P XANES spectroscopy agree with the determined sequential P fractions. In particular, both of the arable soils, A1 and A2, and the sediments S1 to S4 were clearly dominated by Ca-P compounds, according to the XANES spectroscopy (Figure 2). In compliance with this, in these samples, $H_2SO_4$-$P_{mr}$ accounted for the greatest proportion of $P_t$ in the sequential fractionation and this P fraction has previously been characterized as stable, Ca-bound P [26]. The dominant occurrence of Ca-P in sediments from the Baltic Sea [21] or nearshore sediments [48] is not uncommon. The percentages of the Ca-P compounds in the agricultural soils with different fertilization treatments ranged between 0% and 21% [10]; however, due to the elevated amounts of $CaCO_3$ in A1 and A2 (Table 2), most likely from liming, and the underlying glacial till, the formation of Ca-bound P in these soils could have been promoted, reaching proportions of up to 92% (Figure 2).

The organic P compounds were assigned in A1, W2 and S5, in significant proportions, by the XANES spectroscopy (Figure 2). The summed proportions of $P_{mu}$ from the sequential fractionation were also high in A1 and W2 among the soil samples and, thereby, agreed with the results of the XANES spectroscopy (Table 3 and Figure 2). Recently, the importance of $P_o$ compounds for sustainable agriculture has been emphasized [49]. The absolute $P_{mu}$ concentrations were highest in S5 and S6 among the sediments determined with sequential fractionation (Table 3), but this could only be confirmed for S5 by XANES spectroscopy because it ascertained 46% $P_o$ compounds in S5 and 0% $P_o$ in S6. An underestimation of $P_o$ compounds by XANES spectroscopy is likely because the XANES spectra of phytic acid are known to lack strong and distinguishing features [50].

The wetland soils act as sinks for P compounds, particularly Al- and Fe-P and $P_o$ compounds (Figure 2), in the first part of the transect, including the arable and wetland soils. In agreement with an earlier study about P forms along a continuum from agricultural fields to lake sediments, which reported significant P losses from field soils but only small amounts of P in the nearshore lake sediment [48], the samples S5 and S6 at the end of the investigated transect at a water depth of 230 cm can also accomplish a sink function for $P_o$ compounds (Figure 2 and Table 3) among the Bodden sediments. This also agrees with an earlier study, in which lost P from agriculture either became available to biota or was deposited in deeper portions of a lake system [48].

Thus, both the wetland soil and the sediments S5 and S6 at the end of the transect can act as sinks for P, with a lower probability of P mobilization into the above water column. Generally, the investigated transect of soils and sediments can be divided into two separated systems. The first system consists of arable soils, followed by wetland soil. Phosphorus compounds are transported from agricultural fields into the wetland by processes such as runoff and soil erosion, they accumulate, and can be conserved in the latter. The second system comprises the Bodden sediments with a similar sink function for P compounds in deeper sediments at the end of the investigated transect. The results from the sequential P fractionation and XANES spectroscopy suggest no great transfer processes of P species from the first system towards the second.

## 5. Conclusions

In a sample set along a transect from arable and wetland soils to aquatic sediments, the methods of sequential P fractionation and P XANES spectroscopy similarly determined a dominance of Ca associated P at the arable soils and Bodden sediments, and high proportions of $P_{mu}/P_o$ at the surface arable soil (A1) and subsurface wetland soil (W2). Thus, both methods complemented each other in delivering comprehensive results concerning the P characterization and sink functions of sample locations along this transect.

The investigation revealed two sinks along the transect from the arable land to adjacent aquatic lagoon sediments. As the wetland soil acts as a semiterrestrial trap for moderately stable P, Al-P and $P_{mu}/P_o$ compounds, it can help to prevent direct transfer of P, e.g., by leaching or runoff during erosion events from the agricultural field to the adjacent Bodden. An intact *Phragmites* wetland can therefore protect the aquatic ecosystem from further eutrophication. Consequently, it is reasonable to preserve existing buffer strips such as *Phragmites* that stand along water bodies or to potentially create new ones in areas where they are not yet abundant. Among the aquatic sediments, we detected $P_{mu}/P_o$ compounds and stable P fractions accumulating at the deepest sample location compared to the sediments from more shallow positions, closer to the coast. Thus, it is likely that deeper basins in the investigated Bodden system act as sinks for stable P forms, which are not directly consumed by aquatic organisms. If this concept can also be applied to aquatic ecosystems that are different from the coastal lagoons in the Baltic Sea, it has to be investigated, if measures such as the deepening of shallow aquatic areas can help to reduce pollution and eutrophication by trapping P in the sediment of deeper basins. As an important new insight, we can conclude that a gradual transformation of redox-sensitive Al/Fe-bound P into redox-insensitive and, thus, more stable Ca-P compounds

with increasing moisture impact, seems to be a general process in North-east German lowlands, irrespective of the observation scale from small- [23,51] over medium-scale (the present study) to large-scale transects [21]. The P-sink formed by stable Ca P-compounds reduces the eutrophication risk for immediately adjacent waters, such as rivers, lakes and the Baltic Sea lagoon systems.

**Supplementary Materials:** The following supporting information can be downloaded at: https://www.mdpi.com/article/10.3390/soilsystems7010015/s1, Figure S1: Map of the location of investigated arable (A) and wetland (W) soils and sediments (S); Figure S2: P *K*-edge XANES spectra of soil (A, W) and sediment (S) samples Table S1: R factors of the results of linear combination fitting from P *K*-edge XANES analyses of soil (A, W) and sediment (S) samples.

**Author Contributions:** Conceptualization, J.P. and P.L.; methodology, J.P., R.S. and W.K.; software, J.P. and W.K.; validation, P.L. and W.K.; formal analysis, J.P. and R.S.; investigation, J.P. and P.L.; resources, J.P.; data curation, J.P.; writing—original draft preparation, J.P.; writing—review and editing, J.P. and P.L.; visualization, J.P.; supervision, P.L.; project administration, P.L.; funding acquisition, P.L. All authors have read and agreed to the published version of the manuscript.

**Funding:** This research was funded by the Leibniz Association within the scope of the Leibniz ScienceCampus "Phosphorus Research Rostock".

**Institutional Review Board Statement:** Not applicable.

**Informed Consent Statement:** Not applicable.

**Data Availability Statement:** The data that support the findings of this study are available from the corresponding author upon reasonable request.

**Acknowledgments:** The authors thank Elena Heilmann and Britta Balz (Soil Science, University of Rostock) for analytical help concerning P fractionation and ICP measurement. Furthermore, we would like to thank Volker Reiff (Applied Ecology and Phycology, University of Rostock), who was a great support during the sediment sampling. Finally, we are grateful to Jörg Prietzel (Department of Soil Science, Technical University of Munich) for providing the P XANES reference spectra and to the technical staff at BL8 of the SLRI, Thailand for their support during XANES research.

**Conflicts of Interest:** The authors declare no conflict of interest.

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
