# Peer review of "Characterization of Phosphate Compounds along a Catena from Arable and Wetland Soil to Sediments in a Baltic Sea lagoon"

_soilsystems, doi:10.3390/soilsystems7010015_

Round 1

Reviewer 1 Report

The manuscript was well written, and minor revisions were recommended. 

Author Response

Reviewer 1:

Biogeochemical cycling of nutrient overlaps with nutrient regulation.
Answer: Yes; thanks for this comment. We deleted “nutrient regulation” since “biogeochemical cycling” is the more general description.

Please see previous comment.
Answer: See answer to previous comment.

Add a citation for the statement.
Answer: We added a suitable reference [6].
[6] Hansen, N.C., Daniel, T.C. Sharpley, A.N., & Lemunyon, J.L. The fate and transport of phosphorus in agricultural systems. Journal of Soil and Water Conservation 2002, 57(6), 408-417).

Speficy what are the in-depth knowledge.
Answer: ... knowledge at the molecular level ... (has been added to text)
It is not clear to the reader, what it is?
Answer: We specified this (medium scale in the range of hundreds of meters).

Did you want to test whether the theory can be applied to the experimental area? If so, please rephrase this sentence to clarify.
Answer: The hypothesis has been specified.

P 2, ln 76-77: “It is not clear, did you mean the width or length or area?”

Answer: This refers to a transect, as we wrote in the previous line. From that it is clear that we do not refer to an area (Compare Cambridge dictionary for the word “transect”: “a line or narrow area along or within which measurements are taken, and items counted, etc. in scientific studies”).

Did you mean 3 depths? And why you consider 3 depths are enough?
Answer: We decided this according to obvious soil horizon arrangement. We do think that for recent processes likely the uppermost sampling depth is most important. Other sampling depth have been included to improve the experimental basis. Nevertheless, as we did very labor-consuming analyses we had to restrict ourselves in the number of samples taken and analyzed.    

Why 700m?
Answer: It is just the distance between first sampling point in the field and a central part of the lagoon waters still shallow enough to do manual sampling from a boat.

Why two or three? Not more? Why you consider two or three subsamples are sufficient to represent the real condition?
Answer: In some cases two in some cases three. We are sure that these are representative for the point of sampling.

This is obscure, what results are considered as similar results? Add the range of these chemical parameters.
Answer: We specified this by adding the ranges of data differences for individual analyses.

For how long?
Answer: “…. until constant weight" (Added in text)

What does this mean?
Answer: modified to "... bound in humic substances, eventually also associated with Fe and Al ....”
Meaning of humic substances is basic knowledge in soil sciences and does not need to be explained in more detail.

Why not a specific number of scans per sample?
Answer: Because of visual inspection of the scan quality. We added this.

Please specify the treatment in the current experiment.
Answer: We added an amendment to the text, explaining what we did statistically.

Reviewer 2 Report

The study involved the determination of phosphate compounds along the catena in arable and wetland soils and the sediments of the Baltic Sea.

  The authors thoroughly described the research facility. In my opinion, although the authors provide the coordinates of the research facility, a map with the research location would be helpful.

The research methodology was adopted correctly and did not raise my doubts. Also, the field measurements themselves (sampling and location) are correct. The research results are also statistically developed—conclusions resulting from the course of research.

The research is interesting and may have practical applications in the case of water protection against nutrients.

Author Response

Thanks to the reviewer for this positive evaluation. There are no critical points to be considered.

Reviewer 3 Report

The research is certainly of international interest. It is original, of particular interest and can certainly stimulate research on this topic. Although this manuscript presents important data to the scientific community, it yet had to be revised for the following minor issues before it can be considered for publication.

1) It is unclear how much samples were analyzed during the study period in different locations? 

2) the discussion section could be improved. There is a lack of comparison with other researchers works. Also, authors should think about the on the use of abbreviations (site locations). now it's hard enough to read and get caught up in the abbreviations provided. you still need to look at the methodological part.

3)In the discussion part, authors should more describe the getted results. Because now sometimes is unclear what significance the dominance of P fractions has for the ecosystem? How could this affect eutrophication?

Author Response

1) It is unclear how much samples were analyzed during the study period in different locations? 

Answer: Considering a similar comment by reviewer 1, we improved the description of sampling and sample pretreatments. We also added a figure showing the sample locations of the transect.  

2) the discussion section could be improved. There is a lack of comparison with other researchers works. Also, authors should think about the on the use of abbreviations (site locations). now it's hard enough to read and get caught up in the abbreviations provided. you still need to look at the methodological part.

We have added some newer references/data sets for comparison. Otherwise we are sure that we have considered the most comparable data sets.

3) In the discussion part, authors should more describe the getted results. Because now sometimes is unclear what significance the dominance of P fractions has for the ecosystem? How could this affect eutrophication?

We made the link between atable Ca-bound P as terminal P form once redox does not play a role anymore, and the reduced risk for eutrophication arising from the stable Ca P-compounds in discussion and revised conclusions.